# The Influence of Polishing and Artificial Aging on BioMed Amber^®^ Resin’s Mechanical Properties

**DOI:** 10.3390/jfb14050254

**Published:** 2023-05-02

**Authors:** Anna Paradowska-Stolarz, Marcin Mikulewicz, Mieszko Wieckiewicz, Joanna Wezgowiec

**Affiliations:** 1Division of Dentofacial Anomalies, Department of Orthodontics and Dentofacial Orthopedics, Wrocław Medical University, Krakowska 26, 50-425 Wrocław, Poland; 2Department of Experimental Dentistry, Wrocław Medical University, Krakowska 26, 50-425 Wrocław, Poland

**Keywords:** dental resins, compression test, tensile modulus, 3D print, dentistry

## Abstract

Currently, 3D print is becoming more common in all branches of medicine, including dentistry. Some novel resins, such as BioMed Amber (Formlabs), are used and incorporated to more advanced techniques. The aims of the study were to check whether or not polishing and/or artificial aging influences the properties of the 3D-printed resin. A total of 240 specimens of BioMed Resin were printed. Two shapes (rectangular and dumbbell) were prepared. Of each shape, 120 specimens were divided into four groups each (with no influence, after polishing only, after artificial aging only, and after both polishing and artificial aging). Artificial aging took place in water at the temperature of 37 °C for 90 days. For testing, the universal testing machine (Z10-X700, AML Instruments, Lincoln, UK) was used. The axial compression was performed with the speed of 1mm/min. The tensile modulus was measured with the constant speed of 5 mm/min. The highest resistance to compression and tensile test were observed in the specimens that were neither polished nor aged (0.88 ± 0.03 and 2.88 ± 0.26, respectively). The lowest resistance to compression was observed in the specimens that were not polished, but aged (0.70 ± 0.02). The lowest results of the tensile test were observed when specimens were both polished and aged (2.05 ± 0.28). Both polishing and artificial aging weakened the mechanical properties of the BioMed Amber resin. The compressive modulus changed much with or without polishing. The tensile modulus differed in specimens that were either polished or aged. The application of both did not change the properties when compared to the polished or aged probes only.

## 1. Introduction

The future use of 3D print technology is becoming the truth of today’s life. This technology led to the high development of today’s medicine. This phenomenon also extends to dentistry. Although first introduced to restorative dentistry (including simple, conservative restorations, as well as prosthetic appliances) and surgery, the use of this type of material broadened over time. Today, they are preferably used to produce both surgical guides for implantology and craniofacial surgery, but also the basic models [1]. Recently, 3D print was introduced to orthodontics, endodontics, and periodontology as well. It is preferably used for the fabrication of precise appliances, especially when the skeletal anchorage is planned [2,3]. Although the properties of the 3D-printed materials and traditional ones tend to be nearly the same, the printed ones, due to the process of preparation, characterize higher accuracy in shape and time needed for production decrease. Unfortunately, this solution is far more expensive than the traditional one, and therefore, might be unprofitable [4]. Lately, more advanced, 4D technologies were introduced and presented as more precise as well as preferably used, and the difference between the 3D and 4D printing lies in the layer-by-layer print of the specimen. Although the 4D print was not introduced into the treatment yet, as it is still in the phase of experimentation, it gives potential chances for further development of the discipline of biomaterial preparation [5].

The 3D materials are preferably tested by the researchers [6,7,8,9]. For this reason, the different material properties are measured. Although the search for perfect materials is the main focus of the materials science, not many researchers compare the properties of the materials after subjection to the different external factors. Those factors may influence the properties of the materials, including their stability, resistance, and accuracy. Therefore, we decided to compare the properties of the chosen 3D-printed resin to polishing and artificial aging.

BioMed Amber is, according to the producer, a biocompatible material for short-term use. The material is strong and rigid. It is transparent, but has a yellowish glow, so it is not truly esthetic. The material is also durable with the frequent use of common disinfection agents and sterilization [6,7].

The use of materials in the oral cavity exposes them to unfavorable conditions. The factors influencing the materials depend on applications of the prepared item. One of the tests that is performed is artificial aging, which should mimic the oral cavity conditions [10]. Two main methods of artificial aging are possible: thermocycling and the use of water or artificial saliva immersion in time [11,12]. For this reason, we decided to subject the chosen material to unfavorable conditions. We chose the prolonged storage in water at 37 °C and compared the resin blocks in compression and tensile modulus tests. We also decided to check whether the mechanical features changed due to the polishing of the materials, as this process is necessary in each material preparation to reduce its roughness and improve aesthetics. To the best of the authors’ knowledge, this is the first time this kind of research with this specific resin was conducted. Moreover, the tests are often prepared on the blocks of resins that find their uses in the preparation of dental restorations. The presented study could help us find another potential use of the presented resin.

The aim of this study was to check if the mechanical purposes of the selected resin differed after its polishing. Another goal was to check whether artificial aging influences the mentioned properties. The third aim was to find the correlation between artificial aging and polishing when it comes to the resistance to compression and tensile tests.

When planning the research, the hypotheses were formed:1.There was no influence of polishing on the properties of the examined resin;2.There was no influence of artificial aging on the properties of the examined resin;3.There was no influence when both polishing and artificial aging were applied on the properties of the examined resins;4.No differences were observed between the polished and aged samples.

## 2. Materials and Methodology

### 2.1. Materials

A 3D-printed resin, BioMed Amber (Amber UFI number E300-P0FU, Formlabs Ohio, Millbury, OH, USA), was evaluated in this study. The material is biocompatible and characterizes with various properties and potential uses. Recommendations regarding its selected applications, provided by the manufacturer, are summarized in Table 1.

### 2.2. Specimens’ Preparation and Artificial Aging

Two types of specimens were used for the evaluation of the selected properties: rectangular ones (for a compression test in accordance with the ISO 604:2003 standard) and dumbbell-shaped specimens (type 1BA) (for a tensile test in accordance with the ISO 527-1:2019(E) standard) [13,14]. The number of specimens printed for each test was n = 120 of rectangular specimens and n = 120 of dumbbell-shaped specimens. According to ISO standard, the appropriate number of samples for this kind of research is 5, but the authors considered this number too small and expanded the sample size to 30 for each test. After that, the prepared samples were divided into four groups (30 specimens each), to be tested.

The specimens were printed using a Form 2 printer (Formlabs), following the mentioned ISO standards and the producers’ instructions. The printer is self-adjusting, and the printing properties are set once the cartridge is in the printer. The Form 2 printer (Formlabs) has a violet light (405 nm) and the power of 250 mW. After printing, the specimens were rinsed 2 × 10 min in 99% isopropanol alcohol (Stanlab, Lublin, Poland). Then, the specimens were dried at room temperature for 30 min and post-cured in Form Cure (Formlabs). The following settings of post-curing were applied: 30 min in 60 °C. Finally, the supports were removed, and the specimens were grinded with sandpaper. One half of the specimens (n = 60 of rectangular specimens and n = 60 of dumbbell-shaped specimens) were polished with a 0.2 pumice (Everall7, Warsaw, Poland) and polishing paste (Everall7) on one side of the specimen.

Afterwards, the specimens were stored at room temperature and 50% humidity for 24 h (before a tensile test) or for 4 days (before a compression test) and one half of the specimens were subjected to the tests. The other half of the specimens were artificially aged for 90 days in distilled water at 37 °C before testing (the water was changed every week).

### 2.3. Compression Test

The dimensions of the specimens ((10.0 ± 0.2) mm × (10.0 ± 0.2) mm × (4 ± 0.2)) in mm were selected to meet the requirements specified in the ISO 604:2003 standard. Before the test, the specimens were conditioned in the air at 23 °C/50% RH for 4 days. Afterwards, the width and the height of the specimens were measured in five points, using a Magnusson digital caliper (150 mm) (Limit, Wroclaw, Poland), and then, the mean values were calculated. The axial compression test was performed using the universal testing machine (Z10-X700, AML Instruments, Lincoln, UK) at the constant speed of 1 mm/min. The tested sample in the testing machine is presented in Figure 1. The measurements performed allowed for the determination of the compressive modulus (*E* [MPa]) of each specimen, as a slope of a uniaxial stress–strain curve recorded, based on the calculation of:(a) compression stress (*σ* = F:A [MPa]),
where: F—force [N], A—initial cross-sectional area measurement [mm^2^],
(b) nominal strain (*ε* = ΔL:L),
where: L—the initial distance between the compression plates [mm], ΔL—the decrease in the distance between the plates after the test [mm]).

Additionally, the dimensions of each specimen were measured both before and after the compression in order to enable evaluation of the changes in width and height due to compression.

### 2.4. Tensile Test

The dumbbell-shaped specimens (type 1BA) with the length of 75 mm, width of 10 mm, and thickness of 2 mm were printed, following the ISO 527-2:2019 norm. Before the test, the specimens were conditioned in the air at 23 °C/50% RH for 24 h. The width and the height of the specimens at the test length were measured in five points, using a Magnusson digital caliper (150 mm) (Limit, Wroclaw, Poland), and then the mean values were calculated. The tensile test was performed using the Universal Testing Machine (Z10-X700, AML Instruments, Lincoln, UK) at the constant test speed of 5 mm/min (Figure 2).

The specimens that broke outside of the test length were disclosed. Based on the measurements performed, the stress and strain for each specimen were determined as:(a) tensile stress (*σ* = F:A [MPa]),
where F—force [N], A—initial cross-sectional area measurement [mm^2^],
(b) nominal strain (*ε* = ΔL:L),
where L—the initial distance between the grips [mm], ΔL—the increase in the distance between the grips after the test [mm]).

These calculations allowed for the determination of the tensile modulus (*E*_t_) of each specimen, as:E_1_ = *σ*_2_ − *σ*_1_*ε*_2_ − *ε*_1_ [MPa],
where *σ*_1_ is the stress, expressed in megapascals [MPa], measured at the strain value *ε*_1_ = 0.0005; *σ*_2_ is the stress, expressed in megapascals [MPa], measured at the strain value *ε*_2_ = 0.0025.

### 2.5. Statistical Analysis

All statistical data were prepared using the program Statistica v. 13 (TIBCO Software Inc., Palo Alto, CA, USA).

Mean values with standard deviation of the compressive and tensile modulus of the examined probes were determined. Afterwards, the Kruskal–Wallis test by ranks was stated to check the potential statistical differences between the four examined probes. The p value was set for *p* < 0.001 for this test. Afterwards, multivariate analysis of the variance (MANOVA) test was performed to compare the obtained results.

## 3. Results

After the tests were performed, the authors of this study obtained the following results. In Table 2, we present the compressive and tensile modulus of the prepared probes. The number lower than n = 30 samples means that the specimens broke outside the tested area. For this reason, the authors excluded defective elements and they were not presented in the study. As one could observe, only two polished probes without artificial aging did not manage the tensile test. The compression and tensile tests were performed in four groups for each test (group 1—no polishing and artificial aging, group 2—no polishing, after artificial aging, group 3—after polishing, no artificial aging, and group 4—after both polishing and artificial aging).

In Figure 3 and Figure 4, we present the elasticity module when compression (Figure 3) and tensile modulus (Figure 4) were tested. The materials were divided into four groups and assigned as: A0P0—without being subjected to polishing or artificial aging, A0P1—after polishing, but without artificial aging, A1P0—after artificial aging, but without polishing, and A1P1—after both artificial aging and polishing.

The results indicate that artificial aging strongly weakens the specimens at the compression test, no matter if they were polished or not. The polishing itself does not statistically influence the resistance to compression.

Some similar results are observed in the tensile test (Figure 4). Artificial aging weakens the specimens and the tensile tension measurements are lower. The polishing itself does not statistically influence the resistance to the tensile test. In the aged group, polishing did not change the resistance to the tensile test significantly.

The MANOVA test was applied to find the correlation between the artificial aging and polishing of the specimens. In Table 3 and Table 4, as well as in Figure 5, the MANOVA test for elasticity modules were presented to establish the influence of artificial aging and polishing after the compression (*E_c_*) and tensile (*E_t_*) tests. The mean value of the elasticity module at compression was significantly influenced by artificial aging alone and the interaction of artificial aging and polishing. Polishing alone did not influence the properties of the resin. The mean value of the tensile modulus was strongly influenced by separated procedures of polishing or artificial aging. When both of the technological treatments were applied, no statistically significant difference was observed in comparison to the artificial aging without polishing.

As observed in Figure 5, in the comparison of the elasticity modulus to compression (*E_c_*), statistically significant interaction between artificial aging and polishing was observed. Among the specimens that were not treated with artificial aging, polishing alone lowered elasticity modulus significantly. Among the specimens that were aged, polishing increased the compressive modulus. No interactions were presented when the tensile modulus (*E_t_*) was compared.

## 4. Discussion

The 3D-printable materials are the main focus of attention for today’s medicine, including dentistry. In this field, the use is not limited to prosthetics and surgery anymore, but expanded to orthodontics, endodontics, and periodontology; therefore, new materials are investigated. They are tested in different conditions to find their most proper use. BioMed Amber is a quite new resin that was not yet thoroughly investigated. Due to potential use in preparation of surgical guides (especially for implants and orthodontic miniimplants), the precision of their preparation is crucial [6,7]. Without this type of resin, the preparation of guides for implant insertion would not be possible. Preparation of surgical guides increases the cost of implantation, but they help to obtain the exact point of implantation, especially that the surgical procedure is planned based on the CBCT image [4,15].

The research presented by us was designed based on the ISO standards, which assumes that the number of samples needed for the examination of the presented mechanical features is 5 [13,14]. The comparative studies that checked the properties of mechanical features of the BioMed Amber were conducted on the 10 samples [6,7]. The authors of the presented research focused on bringing up the novel properties of the presented resin, possibly broadening its future use. Therefore, we decided for artificial aging, simulating the intraoral condition. Thermocycling is another method of aging for the specimens, which is mainly used when restorative materials and cements are taken into account [16]. Thermocycling, in addition, predicts the fact that the material is used in the patient’s mouth when the changes in the temperature are applied (e.g., while eating and drinking). The resin we tested is transparent and yellowish, which means it would not be used for restorative dentistry; therefore, we did not consider thermocycling as the method of testing. When the research was planned, we decided to expand the specimens to thirty pieces for each test. Therefore, the authors think it is a good comparison result to present the points that were inspected.

Most of the resins that are tested before and after the artificial aging are compared in terms of the color stability [17,18,19,20], which should not be the only factor taken into consideration when material is examined. An interesting study, which should be mentioned in the discussion, was published by Mazzitelli et al. [21]—the researchers presented the results that the chemical composition of the material influences the color stability. During the materials preparation, different processes are performed; one of them is polishing. This could influence the material’s surface. Moreover, the use of beverages and food as well as bleaching, damages the surface of the materials applied in dentistry [22,23]. The color change is also observed when cigarette smoke is an influencing factor [24]. When considering artificial aging, besides the esthetic agonizing, other properties, such as tensile modulus and compressive modulus, are rarely taken into account. Therefore, the results presented in our paper might be one of the few available papers on that topic. We showed that artificial aging negatively influences the examined resins, which is also consistent with other material properties. Some new, more resistant materials are being searched for; however, most of the attention of the producers focuses on the composite blocks used for teeth restoration [25]. Another interesting study [26] showed the shear bond strength of materials used for the fabrication of occlusal splints, which shows that not only 3D-printed materials are influenced by the artificial aging procedures.

The wear of the material depends on the properties of the material itself [27]. In addition to the resins, other materials (metals, composites, ceramics, and other polymers) are incorporated into use [28]. In our study, we focused on the 3D resin that is used for short-term mucosal and tissue contact in medicine. The studies performed on composites [29] reveal that polishing lowers the protective properties of the materials, but we know that it is impossible to use the dental filling without the softening of the material’s surface; therefore, polishing is an inseparable process in material preparation for its use. In the presented study we used a transparent, rigid material that is widely used in dentistry. The presented paper is one in the series of papers regarding BioMed Amber resin; the authors focused on searching for novel properties of the presented material, which could expand its future use. By storing the samples in water, the authors showed that possible long-term use in the oral cavity would be influenced by the wet environment. We showed that in the case of BioMed Amber resin, polishing reduces the resistance of the material to compression and tension, which is also consistent with the results of other studies on dental fillers [29,30,31]. This is consistent with a theorem that material gets weakened after polishing. These results may correspond also to reported changes in fractal dimension (FD) and texture analyses (TA) presented in previous studies. The study showed that Amber resin shows a high number of changes in the fractal and texture analyses [32]. The presented study referred to the compression test only, with no influence of external conditions.

The weakest point of this research is the fact that only a selected resin was taken into account. However, due to the lack of content on that type of research, the authors had to plan the whole paper from the beginning, basing the research on referring to the composites. In addition, most of the published papers, which concentrate on the topic of dental materials (such as the one planned by us), do not refer to compression and tensile modulus. They also place more attention on the restorative materials. These limitations make the discussion regarding that topic more difficult. These limitations, though, make us want to expand our research to other potential resins.

## 5. Conclusions

The presented study allowed us to form several conclusions. Both polishing and artificial aging change the properties of BioMed Amber resin. Artificial aging changes the compressive modulus of the selected resin, no matter if the specimen was polished or not. The tensile modulus differs when either polishing or artificial aging were conducted. The application of both artificial aging and polishing does not change the properties much when compared to artificial aging without polishing. In general, the polishing of the samples may protect them from the influence of artificial aging, although the properties of the samples weaken. Further investigations should concentrate on that type of testing, and they should not be limited to the transparency and esthetics of the potential materials, because both artificial aging and polishing change the properties of the materials.

## Figures and Tables

**Figure 1 jfb-14-00254-f001:**
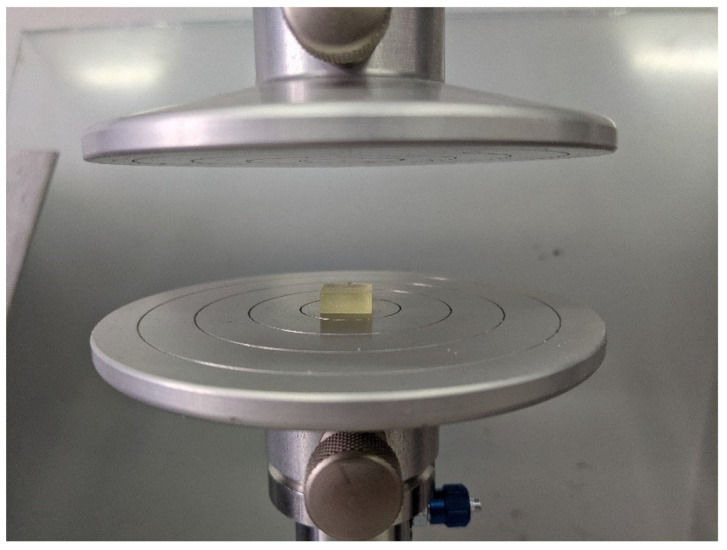
BioMed Amber resin at the compression test.

**Figure 2 jfb-14-00254-f002:**
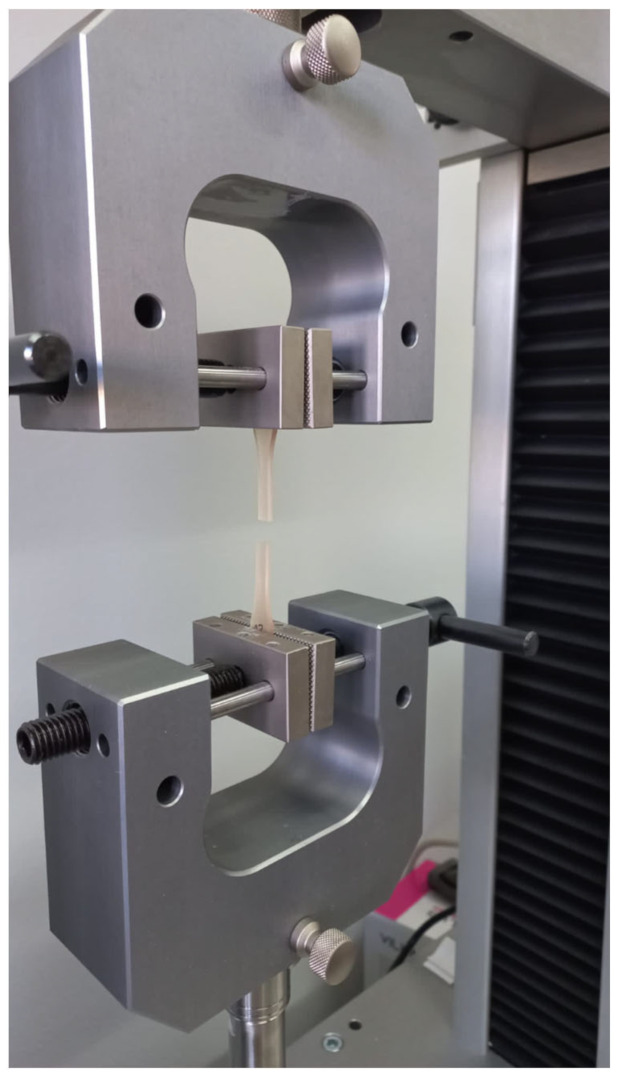
BioMed Amber resin at the tensile test.

**Figure 3 jfb-14-00254-f003:**
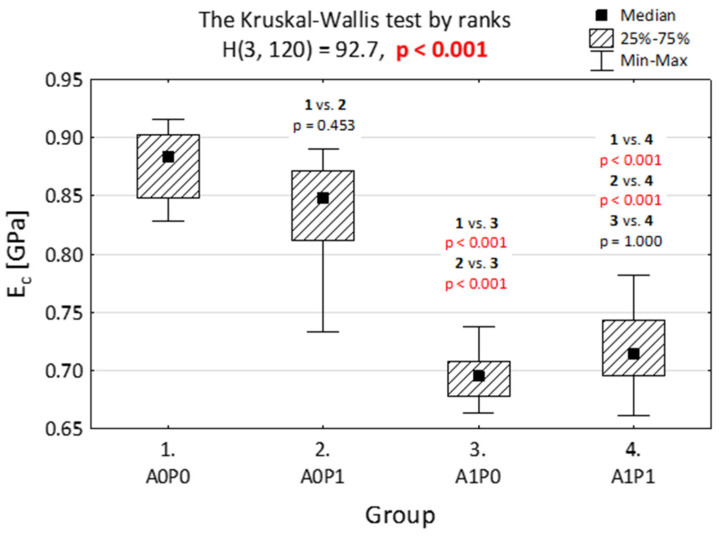
Elasticity module of the compression (*E_c_*) of BioMed Amber when subjected to various technological treatments. Four groups were separated: A0P0—without artificial aging and polishing, A0P1—without artificial aging, but after polishing, A1P0—after artificial aging, without polishing, and A1P1—after both artificial aging and polishing.

**Figure 4 jfb-14-00254-f004:**
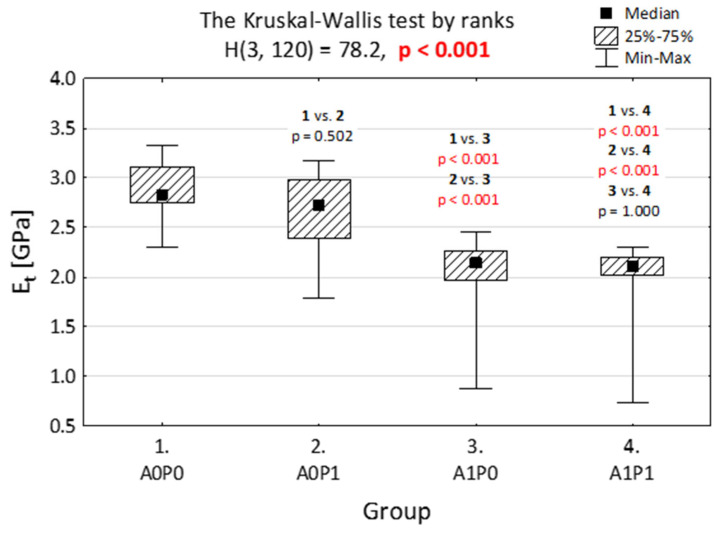
Elasticity module to tension (*E_t_*) of BioMed Amber when subjected to various technological treatments. Four groups were separated: A0P0—without artificial aging and polishing, A0P1—without artificial aging, but after polishing, A1P0—after artificial aging, without polishing, and A1P1—after both artificial aging and polishing.

**Figure 5 jfb-14-00254-f005:**
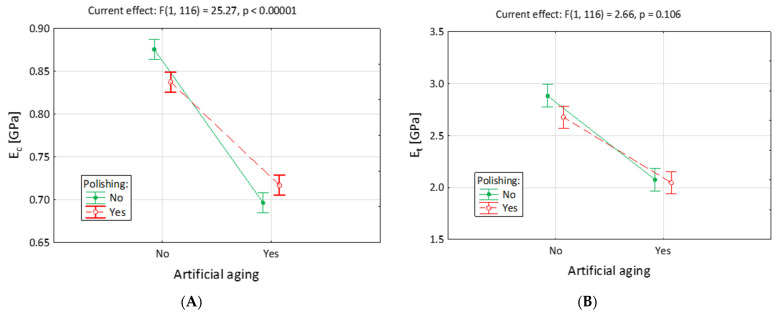
Elasticity modulus under compression (*E_c_*) and tension (*E_t_*) of BioMed Amber resin, after different technological treatments. Subfigure (**A**) refers to compression and (**B**)—to tension. (A0P0—without artificial aging and polishing, A0P1—without artificial aging, but after polishing, A1P0—after artificial aging, without polishing, and A1P1—after both artificial aging and polishing).

**Table 1 jfb-14-00254-t001:** A brief description of the BioMed Amber applications recommended by the producer.

Resin	Application
BioMed Amber Resin	Biocompatible applications requiring short-term skin or mucosal membrane contact suitable for: strong, rigid parts such as functioning threads; end-use medical devices; cut + drill guides (surgical); implant sizing models; specimen collection kits.

**Table 2 jfb-14-00254-t002:** Compressive and tensile modulus of BioMed Amber resin at the compression (E_c_) and tensile (E_t_) tests.

E (GPa)	Polishing	Aging	N	M ± SD	Me [Q1–Q3]	Min–Max
Compression *E_c_*	No	No	30	0.88 ± 0.03	0.88 [0.85–0.90]	0.83–0.92
	No	Yes	30	0.70 ± 0.02	0.70 [0.68–0.71]	0.66–0.74
	Yes	No	31	0.84 ± 0.04	0.85 [0.81–0.87]	0.73–0.89
	Yes	Yes	30	0.72 ± 0.03	0.71 [0.70–0.74]	0.66–0.78
Tensile *E_t_*	No	No	30	2.88 ± 0.26	2.83 [2.74–3.11]	2.30–3.33
	No	Yes	30	2.07 ± 0.30	2.07 [0.88–2.45]	2.30–3.33
	Yes	No	28	2.67 ± 0.36	2.72 [2.39–2.98]	1.78–3.17
	Yes	Yes	30	2.05 ± 0.28	2.11 [2.01–2.19]	0.74–2.30

*M*—mean, *SD*—standard deviation, *Me*—median (50th percentile), *Q*1—lower quartile (25th percentile), *Q*3—upper quartile (75th percentile), *Min*—smallest value, and *Max*—greatest value.

**Table 3 jfb-14-00254-t003:** MANOVA results for elasticity module for compression of BioMed Amber resin.

Effect	SS	df	MS	F	*p*
Constant	73.26	1	73.26	71835	<0.000
Artificial aging	0.674	1	0.674	660.6	<0.000
Polishing	0.002	1	0.002	2.28	0.134
Artificial aging × polishing	0.026	1	0.026	25.3	<0.000
Error	0.118	116			

**Table 4 jfb-14-00254-t004:** MANOVA results for elasticity module for tensile modulus for BioMed Amber resin.

Effect	SS	df	MS	F	*p*
Constant	702.2	1	702.2	7870	<0.000
Artificial aging	15.46	1	15.46	173.3	<0.000
Polishing	0.420	1	0.420	4.711	<0.032
Artificial aging × polishing	0.237	1	0.237	2.656	0.106
Error	10.35	116			

## Data Availability

All detailed data could be found at authors A.P.S. and J.W.

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
