# Peer review of "The Influence of Polishing and Artificial Aging on BioMed Amber® Resin’s Mechanical Properties"

_jfb, 2023, doi:10.3390/jfb14050254_

Round 1
Reviewer 1 Report
The study seems interesting and genuine, however the authors should address the following points in the manuscript:
- The abstract should be non-structured with specific word count (please check the authors' guidelines).
- English revision is required since some sentences are difficult to understand. Also, it is advised to use passive voice for manuscript preparation.
- Please add hypothesis/hypotheses at the end of introduction section.
- How did the authors determine the sample size? please explain.
- Why did the authors consider compression and tensile only? why not fracture strength?
- No information was found on thermal cycling. How did the authors perform aging? why is aging important here if its use if for short term?
- What could be the future recommendations of this study? please add it/them to the manuscript.
- Clarify the clinical importance of this study.
- Summarize the abstract using bullet points.
The manuscript needs minor English editing.
Author Response
Dear Reviewer 1,
thank you for the time and effort spent on the corrections of the manuscript. Please, find our comments in the additional file. Thank you - Authors.

Reviewer 2 Report
The article entitled “The influence of polishing and artificial aging on BioMed Amber® resin’s mechanical properties.” aims to check if the mechanical purposes of the selected resin differed after its polishing. Another goal was to check whether artificial aging influences the mentioned properties. The third aim was to find the correlation between artificial aging and polishing on the resistance to compression and tensile tests
The article covers an interesting topic. Minor changes are required.
The authors wrote in line Table 1 that BioMed Amber applications recommended by the producer are “Biocompatible applications requiring short-term skin or mucosal membrane contact”. Why the author tested therefore after an aging simulation since it should be a material that should not “age” in the mouth?
Please elaborate and explain in the introduction or in the discussion.
Line 63:
The authors could explain why water aging was preferred to thermocycling
The authors wrote:
“Most of the resins that are tested before and after the artificial aging, are compared in terms of the color stability [16-19], which should not be the only factor taken into consideration when material are examined. “
The authors could cite also reviews from Paolone Gaetano research group on color stability (search name and color stability on pubmed). There are many of them, and since they are systematic reviews they should be considered.
The authors wrote:
“This is consistent with a general theorem, that material gets weakened after polishing. “
There is no space in a scientific article for “general theorems” unless solid references are cited.
Author Response
Dear Reviewer,
thank you for the positive feedback. Our responses are presented in the uploaded file.
Thank you - Authors
